# Precise coupling of the thalamic head-direction system to hippocampal ripples

Guillaume Viejo[1] & Adrien Peyrache [1]✉

The anterior thalamus is a key relay of neuronal signals within the limbic system. During sleep, the occurrence of hippocampal sharp wave-ripples (SWRs), believed to mediate consolidation of explicit memories, is modulated by thalamocortical network activity, yet how information is routed around SWRs and how this communication depends on neuronal dynamics remains unclear. Here, by simultaneously recording ensembles of neurons in the anterior thalamus and local field potentials in the CA1 area of the hippocampus, we show that the head-direction (HD) cells of the anterodorsal nucleus are set in stable directions immediately before SWRs. This response contrasts with other thalamic cells that exhibit diverse couplings to the hippocampus related to their intrinsic dynamics but independent of their anatomical location. Thus, our data suggest a specific and homogeneous contribution of the HD signal to hippocampal activity and a diverse and cell-specific coupling of non-HD neurons.

[1] Montreal Neurological Institute, McGill University, Montreal, QC, Canada. ✉email: adrien.peyrache@mcgill.ca

Thalamocortical loops form the canonical circuits of the most complex cognitive functions[1–4]. The dorsal thalamus (henceforth referred to as the thalamus for simplicity) is organized in a large family of cytoarchitectonically defined nuclei[1]. Among them, the anterior thalamic nuclei (AT) are located at the central stage of the Papez circuit[5] and play a key role in memory[6,7], spatial navigation[8–10], and arousal states[11,12]. However, the dynamics of AT neurons and their functional integration with the limbic system remain elusive.

The role of AT in navigation is indicated by the presence of head-direction (HD) neurons, which discharge when the animal is facing a particular direction and are primarily located in the anterodorsal (AD) nucleus[8,10]. HD cells of the AD are necessary for the establishment of higher level spatial representations in the brain's navigation system, for example the grid cells of the medial entorhinal cortex[9].

The relationship between neuronal activity and memory in the AT remains elusive. In contrast, this relationship has now been clearly established in the hippocampus. Non-Rapid Eye Movement (NREM) sleep is instrumental for memory consolidation[13]. During NREM sleep, the hippocampal activity patterns that form during exploration are reactivated[14], thus recapitulating spatial trajectories in the environment[15,16]. These replay events are associated with SWRs in local field potential (LFP)[17] and are necessary for memory[18–20]. However, it remains unclear whether this phenomenon is entirely generated within the hippocampus or is influenced by its inputs.

During NREM sleep, the timing of SWRs is orchestrated by thalamocortical dynamics[21–24], especially the thalamocortical slow oscillation that arises from the fluctuation of membrane potentials between UP and DOWN states (corresponding to active and silent states, respectively)[25]. The AD is a key hub for the propagation of the slow oscillation[12], suggesting that HD cells play a role in coordinating neuronal activity within the limbic system during NREM sleep.

The contribution of AD HD cells to offline processing of spatial signals is further demonstrated by the fact that HD cells maintain a coherent firing activity such that a single direction can be decoded at any moment during sleep[10,26]. Additionally, AD HD cells lead their main cortical targets (in the post-subiculum), independently of brain states[27]. This suggests that during sleep AD neurons continue conveying a coherent HD signal to the navigation system, including the hippocampus, independent of the current heading of the animal[28].

The role of the other AT nuclei in coordinating activity in the limbic system and how this activity relates to neuronal dynamics is still unclear. One challenge when investigating thalamic function is that, while each nucleus is characterized by specific connectivity with other brain areas, exact input/output patterns can differ across neurons from the same anatomically defined nucleus[1,29]. Furthermore, the firing of thalamic neurons is often believed to be homogeneous (e.g. burst firing during NREM sleep[25,30]), yet variability in spike train dynamics has been previously reported[31,32].

By simultaneously recording thalamic neuronal activity and hippocampal LFP in freely moving and naturally sleeping mice, we first addressed the question of how AT neurons, and specifically thalamic HD cells, are coupled to SWRs. Then, we examined how the variability in this coupling correlates with intrinsic properties of AT neurons, as determined from spike train dynamics across brain states and timescales. We show that HD cells are specifically coupled to SWRs, increasing their gain and firing coherently for a particular direction during SWRs. Furthermore, HD cells of the AD nucleus (the majority of cells in this nucleus) form a very homogeneous population of neurons, both in terms of their intrinsic dynamics as well as hippocampal

coupling. Non-HD cells offer a striking contrast: their modulation by hippocampal SWRs and intrinsic dynamics are highly diverse and differ even for proximal neurons of the same nucleus.

## Results

**The HD signal is precisely coupled to SWRs.** We performed recordings of the anterior thalamus ($n = 2016$ AT neurons) in freely moving mice in an open environment, while they foraged for food, and in their home cages during sleep periods that preceded and followed exploration. Neurons were classified as HD cells by measuring the modulation of firing rate with respect to the direction of the head of the animal in the horizontal plane ($n = 161$ HD cells; see Methods). The tuning curves of 24 simultaneously recorded HD cells are shown in Fig. 1a.

HD cells of the AT are believed to convey a coherent signal to the parahippocampal system during sleep[10,27,28]. We thus made the prediction that the HD system provides the hippocampus with a specific direction during SWRs in which place cell ensembles replay previously formed patterns associated with spatial trajectories[15,16]. To this end, we simultaneously recorded LFP in the pyramidal layer of the dorsal CA1.

Around the time of SWRs only a subset of HD neurons fired, and these neurons demonstrated a similar preferred HD (Fig. 1b). To better visualize this effect, we first uncovered the topology of population firing using ISOMAP (see Methods)[26]. This nonlinear dimensionality reduction method revealed the embedding of HD cell population vectors, and was computed without knowledge of the behavioral correlates (i.e. the HD tuning curves). During wakefulness, this topology had a ring structure with the relative angular position along the ring clearly mapping the animal's HD (Fig. 1c). While population activity was restricted to the ring during wakefulness (the center of the ring corresponds to "forbidden states"), its intrinsic dimensionality increased during NREM sleep activity, as seen by the scattering of population vector embeddings, in particular at the center of the ring. This apparent violation of the intrinsic topology resulted from the modulation of network activity by the slow oscillation[25,26].

The population activity around the time of SWRs corresponds to trajectories within the projection space. In the three cases shown in Fig. 1b, activity started near the ring's center (corresponding to putative DOWN states) and settled on fixed points in the outer part of the ring (corresponding to "allowed" states during wakefulness) at the time of SWRs.

To quantify population states around SWRs and across sessions, the population vectors at each time bin were expressed in polar coordinates by taking the center of the ring as the reference point. We analyzed data sets that included more than 10 HD neurons recorded together and showed a clear ring topology during wakefulness ($n = 7$ sessions, 3 animals, see Supplementary Fig. 1). This topology was assessed by examining the distance of each point (i.e. a population vector projected on the manifold) to the center (i.e. mean) of the embeddings. During wakefulness, distances were all distributed within a fixed range from the center and, importantly, no point was close to the center (Fig. 1d and Supplementary Fig. 1). This indicates that the embeddings form an annulus-like shape around the center, as expected for a population of HD cells[26]. At time of SWRs, the HD population pointed to near-random directions (Fig. 1e and Supplementary Fig. 1), and the slight bias towards one direction certainly results from the non-homogeneous sampling of preferred HD in the population (due to the limited number of recorded neurons).

For all sessions examined, the radius of the trajectories peaked around times of SWRs compared to baseline (see Methods) (Fig. 1f). Average radius was maximal before SWR

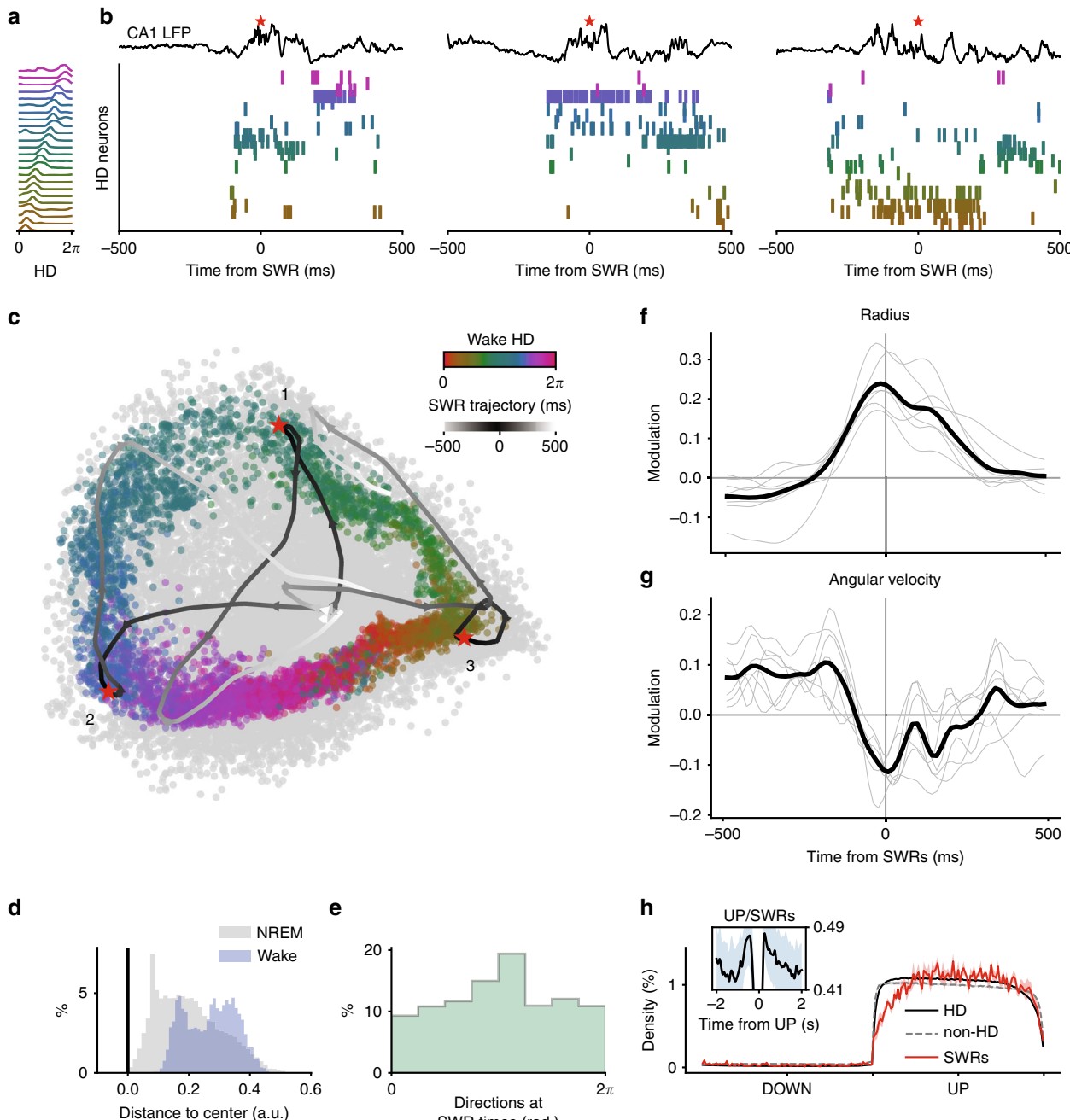

**Fig. 1 Stability of HD decoding before SWRs. a–e** HD decoding with ISOMAP around time of SWRs for one example session. **a** Tuning curves of HD neurons recorded simultaneously in the AD nucleus. **b** Simultaneous recording of CA1 pyramidal layer LFP (top) and neuronal activity in AD (same neurons as in **a**) around the time of SWRs during NREM sleep. Spikes from HD neurons are sorted according to their preferred direction during wakefulness (as in **a**). Red asterisks indicate SWRs. **c** Two-dimensional embeddings (using ISOMAP) of HD neuronal population activity during wakefulness and around the times of SWRs (colored and gray points, respectively). Curves display the population activity embeddings of the three examples shown in **b** (same red asterisks). **d** Distribution of distances from the embedding center (**c**) during wakefulness (blue) and NREM sleep (gray). **e** Distribution of average angular directions around the time of SWRs (±30 ms). **f** Average normalized radii (i.e. distance from embedding centers relative to shuffles) around times of SWRs (gray lines; $n = 7$ sessions, 3 animals) and session-wide average (black line). **g** Same as **f** for angular velocity (i.e. angular displacement between two consecutive activity bins). **h** Average density of spikes (±s.e.m.) for simultaneously recorded HD (black) and non-HD (gray) neurons, and rate of occurrence of SWRs (red line) during DOWN and UP states (normalized time).

peaks ($-17$ ms ± 6 ms; $t = -2.82$, $p = 0.03$; $t$-test). Was the angular position stable or was it drifting at these times? To address this question, we computed the relative angular velocity of the signal (i.e. angular displacement relative to a baseline). Angular velocity was minimal around SWRs (Fig. 1g), and, similarly to the radius, reached its minimum before SWR peaks

($-18$ ms ± 15 ms; $t = -2.88$, $p = 0.02$; $t$-test excluding SWRs occurring <100 ms after UP onset). Similar results were obtained upon using Bayesian decoding of angular position (i.e. a decoding based on HD tuning curves; see Supplementary Fig. 2)[10,33].

It is possible that the stabilization of the network into particular states results from the increase of HD population

firing. However, these two aspects of HD cell population activity seem independent as HD cell population outside of SWRs showed fast sweeps at high population firing rates (i.e. on the outer ring of the population activity topology)[26]. Hence the increase in gain and the stabilization of the network into particular states is characteristic of epochs preceding SWRs.

The coupling between HD neuronal population and SWRs could result from non-specific coordination of SWRs with slow oscillations[22,23,34], in particular at the transition between DOWN and UP states. To test for this possibility, we first detected putative DOWN and UP states from ensemble activity (composed of HD and non-HD neurons) and duration of each epoch was normalized (Fig. 1h). We did not see any change in the probability of SWR occurrence around DOWN/UP transitions of AT activity, but instead saw a constant rate of SWR occurrence during UP states (Fig. 1h). If anything, SWRs were slightly less prevalent during early onset of UP states. In contrast, when the modulation of SWRs by the slow oscillation was examined with a cross-correlation analysis which does not correct for UP state duration, SWRs seemed more prominent at the beginning and end of the UP states (Fig. 1h, inset). However, this analysis primarily reflects the distribution of UP state duration and not a preferred phase within UP states. Finally, we asked whether the co-occurrence of SWRs and stable HD signal was biased by their relative position within UP states. SWRs were clustered according to the time elapsed since the UP state onset. No difference was observed in the time-lags of the radius peaks and minima of angular velocity relative to SWRs. The only exception was for SWRs occurring in the first 200 ms of UP state as, at those times, the system is certainly more governed by dynamics relative to the transition from the DOWN states (Supplementary Fig. 1b, c).

In conclusion, SWRs were specifically preceded by high-activity and stable HD population states independently of the coordination between the thalamocortical slow oscillation and SWRs.

**Homogeneous firing of thalamic HD cells around SWRs.** The observed coupling of the HD network to SWRs raised the question of whether or not this was specific to HD AT neurons. Non-HD AT neurons were recorded at increasing dorsoventral depth from session to session (Fig. 2a), thus spanning most AT nuclei, not only the most dorsal nuclei adjacent to AD. Taking advantage of multiple channels and geometry of silicon probes we were able to reconstruct the putative tracks of each shank across the AT (Fig. 2b). The position of the probes was calibrated along the dorso-ventral axis by the first appearance of spiking activity on at least one shank (indicating the dorsal border of the thalamus) and along the medio-lateral axis by matching the highest density of HD neurons with the AD nucleus[8,10] (Fig. 2b and Supplementary Fig. 3).

As no particular behavioral correlates and no topology of the population activity could be assumed for non-HD neurons, we first analyzed neuronal activation around times of SWRs on a cell-by-cell basis. Similar to the potential caveat of HD cell population analysis (Fig. 1), a challenge when evaluating neuronal responses to SWRs with cross-correlation analysis is that SWRs are co-modulated with the slow oscillation[21–25]. To isolate the specific responses of neurons to SWRs, cross-correlograms were compared to a baseline expected in the condition of an absence of fast co-modulation between neuronal spikes and SWRs (within ±150 ms, corresponding to the typical duration of a sharp wave; see Methods). Cross-correlograms were then expressed in a number of standard deviations ($z$) from these null distributions.

Three example neurons (including one HD neuron) showed different modulation by SWRs (Fig. 2c). The average SWR

modulation of HD neurons shows a clear population response with a highly synchronized increase of firing before the SWRs (Fig. 2d), thus confirming the specific coupling of the HD signal to the SWRs (Fig. 1). On the contrary, non-HD cells showed various responses around the time of SWRs which resulted in a near-uniform average modulation. While this observation suggests that unlike HD cells, non-HD cells do not act as a synchronized population immediately before SWRs, the non-HD cell populations were recorded from different nuclei of the AT (Fig. 2b). It is thus possible that, locally, some nuclei show the same coupling to SWRs as the HD neurons of AD. To rule out this possibility, we computed pairwise correlations of neurons recorded on the same shanks, assuming that those neurons were most likely to come from the same nuclei. As shown in Fig. 2e, in a session containing HD and non-HD neurons, SWR modulation of non-HD neurons was more heterogeneously distributed than that of HD cells. This locally non-homogeneous response of non-HD cell population to SWRs compared to HD neurons was confirmed by computing the distribution of pairwise correlations within all shanks and across sessions (Fig. 2f; $n = 7234$ pairs, 58 sessions, $t = -18.08$, $p = 10^{-71}$, $t$-test).

It can thus be concluded that HD neurons (and, by extension, neurons of the AD nucleus) fire homogeneously around the time of SWRs, pointing in a particular direction. On the contrary, the activation of other AT neurons around the time of SWRs is highly variable, even within local networks (presumably within a nucleus).

**Modulation by the hippocampus is brain-state invariant.** What is the origin of the high variability in the coordination of AT neurons with SWRs? These interactions certainly depend on the input/output connections of each neuron. In the case of a hard-wired network, we hypothesized that the coordination of AT neurons with hippocampal activity should not depend on brain states. During wakefulness and REM sleep, the hippocampus is dominated by theta oscillations (6–9 Hz)[35] which modulate neurons in the entire limbic system[35], including in the AT[36]. We thus tested for a relationship between SWR modulation (during NREM sleep) and phase coupling to theta oscillation (during wakefulness and REM sleep) (Fig. 3a).

The firing of AT neurons relative to REM theta was analyzed, as commonly done, in terms of preferred phase and modulation amplitude. Three example AT neurons, recorded simultaneously but on different shanks of the probe, showed different theta modulation profiles, both in amplitude and preferred phase (Fig. 3b). Overall, 38% of all AT neurons were significantly modulated by theta rhythm during REM sleep ($n = 767/2016$, $p < 0.001$). This proportion was lower during wakefulness ($n = 333/2016$, $p < 0.001$).

How can theta modulation be compared to SWR responses? We first quantified the "SWR energy", defined as the variance of the normalized cross-correlograms (as in Fig. 2). SWR energy and theta modulation amplitude were correlated for the ensemble of AT neurons (Fig. 3c; $r = 0.81$, $p < 0.014$, Pearson's correlation). Those similarities in modulation strength by hippocampal dynamics across different brain states possibly reflect the strength of inputs to each AT neuron from the parahippocampal area, potentially through multiple synaptic pathways. If so, this should also be reflected in the temporal response of each neuron.

Interestingly, HD neurons were overall not modulated by theta (only $n = 21/161$ HD neurons), thus showing once again that HD neurons of the AD nucleus form a homogeneous population of neurons and certainly share the same input/output connectivity profile. Furthermore, the anatomical density of theta-modulated neurons revealed a clear segmentation: they can be found

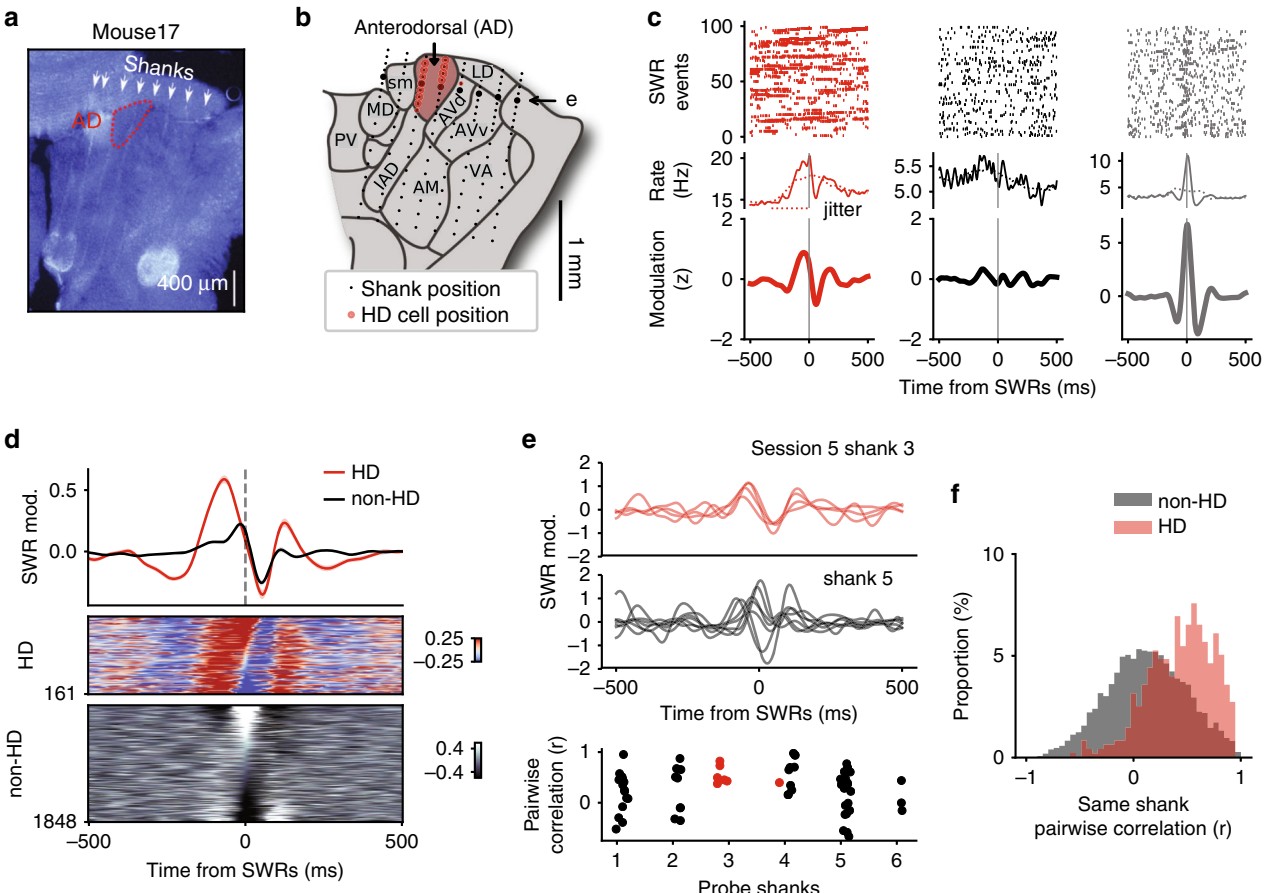

**Fig. 2 Homogeneous coupling of HD, but not non-HD neurons to SWRs. a** Histology of an example mouse (Mouse 17, DAPI staining). Note the tracks of the eight-shank silicon probe and lesion sites at the end of the tracks. Contour of the AD nucleus is shown in red. **b** Schematic of the AT (coronal plane) with recording sites (black dots) for the same animal as in **a**. Red dots indicate where HD neurons were detected. **c** Top, spikes for successive SWRs. Middle, mean firing rates. Bottom, z-scored modulation (see text) for one HD neuron (red) and 2 non-HD neurons (gray) differently modulated by SWRs. **d** Top, average SWRs modulation (±s.e.m.) for HD (red) and non-HD neurons (black) for all sessions. Middle, bottom, SWR cross-correlograms for HD neurons (top) and all other AT neurons (bottom). Neurons were sorted according to the level of correlation with SWR at zero-time lag. **e** Examples of SWR cross-correlograms for HD (top) and non-HD neurons (middle) recorded during the same session but on different shanks (arrow in **b**). Bottom, Pearson's correlation coefficients between SWR cross-correlograms of pairs of neurons recorded on the same shanks. Red dots indicate pairs of HD neurons. **f** Distribution of Pearson's correlation coefficients between SWR cross-correlograms of pairs of HD (red) and non-HD (gray) neurons (similar to **e**) recorded on the same shanks, for all sessions. AD anterodorsal, AM anteromedial, AVd anteroventral, dorsomedial part, AVv anteroventral, ventrolateral part, IAD interanterodorsal, LD laterodorsal, MD mediodorsal, PV paraventricular, sm stria medullaris, VA Ventral anterior.

anywhere in AT except in the putative location of AD (Fig. 3d, see Supplementary Fig. 2 for the three other mice).

Firing of AT neurons relative to SWRs cannot, overall, be trivially described (e.g. excited or inhibited). Rather, they show a wide range of temporal profiles (Fig. 3e for theta-modulated neurons). While modulation by oscillations is commonly characterized in terms of phase preference, such description for modulation by SWRs is lacking. To capture the dynamics of AT neurons around time of SWRs, we used jPCA, a method that captures the rotational dynamics of a neuronal population during non-periodic behavior[37]). The projection onto the jPC subspace describes the various temporal responses of a population of neurons in a two-dimensional trajectory during a pseudo-cycle. Specifically, we determined the jPC basis from the ensemble of z-scored cross-correlograms relative to SWRs (see Methods) and projected each cross-correlogram on the first two jPC components (Fig. 3f). In the two-dimensional jPC projection space, each neuron can be attributed a "phase" that corresponds to the angle from the positive direction on the first jPC axis.

The resulting phases were, for four example neurons, in good agreement with their preferred phase to theta oscillations

(Fig. 3g, note that arrows point towards similar directions for theta and SWR phases). In the population of AT neurons that were significantly modulated by theta, theta phases were correlated with SWR phases (Fig. 3h; $r = 0.18$, $p = 2.3 \times 10^{-7}$, circular correlation). Hence, AT neurons were similarly modulated both in amplitude and in time by hippocampal population dynamics in all brain states, revealing an invariant property at the circuit level.

**A link between firing dynamics and hippocampal modulation.** Although the relationship between the modulation of individual AT neurons by hippocampal dynamics and their detailed connectivity profile is intractable in vivo, does this modulation depend on other intrinsic neuronal characteristics available from extracellular recordings? Spike train dynamics, which are well captured by auto-correlation functions (or "auto-correlograms"), reflect the complex interaction between morphological, input, and membrane properties. Individual HD neurons exhibit quantitatively different auto-correlograms during wake, REM sleep and NREM sleep. Yet, as shown for two example HD neurons, both

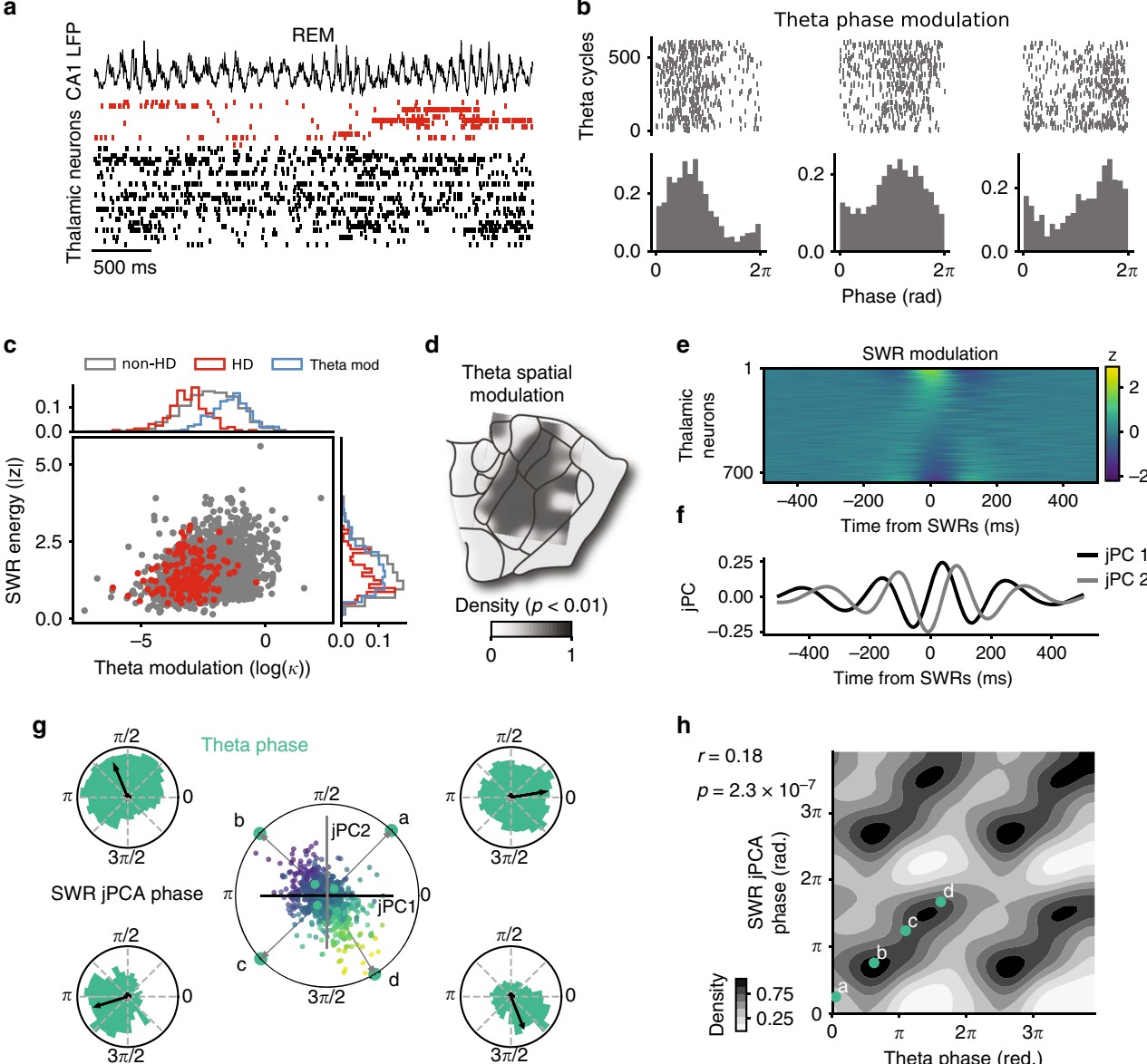

**Fig. 3 Relationship between temporal coupling of AT neurons to SWRs and modulation by theta oscillations. a** Example of a simultaneous recording of LFP in the CA1 pyr. layer (top) and neuronal activity in the AT (bottom) during REM sleep. Red ticks indicate spikes from HD neurons, sorted according to their preferred direction during wakefulness. **b** Top, spikes for successive theta cycles. Bottom, histograms of spike density within theta cycles for three example neurons during REM sleep. **c** Theta modulation versus SWR energy for all neurons (red points indicate HD neurons). Side panels indicate distribution of each group. "Theta mod" (blue distributions) indicates the group of neurons significantly modulated by theta ($p < 0.01$). **d** Density map of theta-modulated neurons (min = 0.016) for one mouse. Note the absence of theta modulation for AD. **e** Z-scored SWR cross-correlograms for all theta-modulated AT neurons (equivalent to the group theta-mod in **c**). Neurons are sorted according to their modulation at SWR peaks. **f** First two jPC vectors (black and gray) of SWR cross-correlograms. **g** Projection of individual SWR cross-correlograms onto the first two jPCs. Point colors indicate 0-lag modulation (same color code as in **e**). Gray arrows indicate corresponding SWR-jPCA phase for four example neurons (green points, a-d). Theta phase distributions for the same four neurons are shown at each corner. **h** SWR-jPCA phase as a function of theta phase ($r = 0.18$, $p < 0.001$, circular correlation). The four example neurons are shown in green.

responses to SWRs and brain state-specific auto-correlograms are similar between HD neurons (only their firing rates, and thus absolute levels of auto-correlation, are different; Fig. 4a). Examination of three non-HD example neurons recorded simultaneously and on the same shank illustrates that nearby neurons can share common properties but can also be strikingly different (Fig. 4b, c). Specifically, a pair of simultaneously recorded neurons from a given shank had similar auto-correlograms and SWR cross-correlograms (Fig. 4b). On the same shank, a third nearby

neuron (see anatomical distribution of waveforms) showed largely different spike train dynamics and opposite modulation by SWRs (Fig. 4c).

The previous examples suggest a high variability in both spike train dynamics and responses to SWRs, even within a given nucleus (with AD being the sole exception). Could these two properties of neuronal firing be related to each other? First, we asked the question of whether neurons showing the same response to SWRs across sessions (and, thus, anatomical

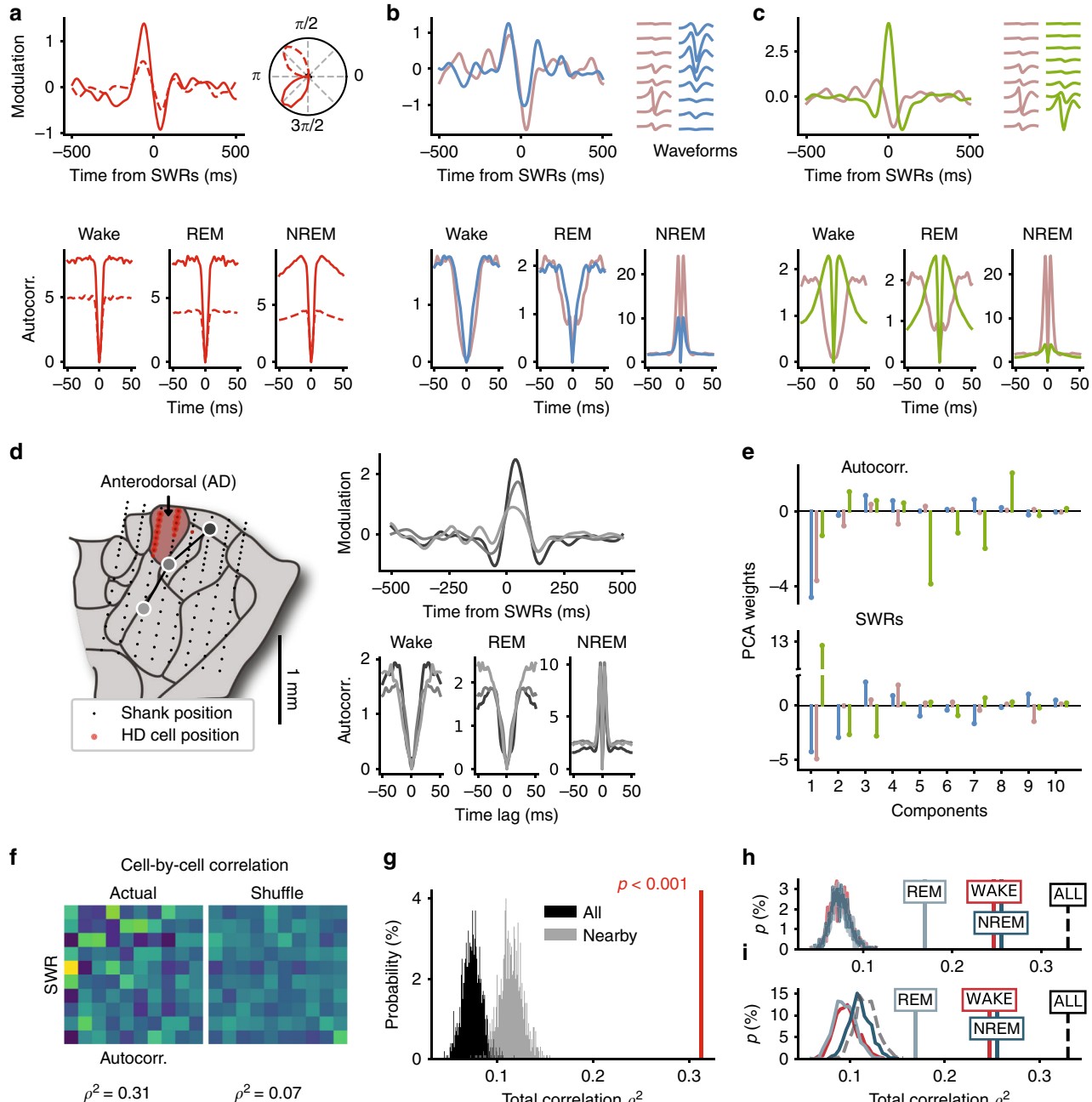

**Fig. 4 Covariation between single-cell spike train auto-correlograms and SWR cross-correlograms. a** SWRs cross-correlograms (top left) and auto-correlograms across brain states (bottom) for two example HD neurons. Circular plots (top right) indicate HD tuning curves. **b**, **c** Same as **a** for three non-HD neurons recorded simultaneously on the same shank. Top right, average waveforms on the eight recording sites of the shank, from dorsal (top) to ventral (bottom) positions. The neuron shown in gray is the same for both panels. Note that, although the cell bodies of the two neurons in **c** are close to each other (see waveforms), auto-correlograms and responses to SWRs are strikingly different. **d** Same as **a**–**c** but for three neurons recorded on different shanks and different sessions in the same animal. Their anatomical location is shown on the left-side map of the anterior thalamus. **e** PCA projections of first 20ms of auto-correlograms (top) and SWR cross-correlograms (bottom) for the three example neurons of **b**, **c**. **f** Correlation matrix between the two series of PCA weights (left) and for shuffled neuronal identities (right) for all neurons. $\rho^2$: total correlation. **g** Distribution of total correlation $\rho^2$ for actual (red line) and for random shuffling of cell identities (1000 times) between all cells irrespective of shank and recording days (black) and between cells recorded on the same shank on a given day (gray). **h** Distribution of total correlation $\rho^2$ for actual (vertical lines) and shuffled data between SWRs and auto-correlograms from wake (red), REM sleep (light blue) and NREM sleep (dark blue). Dashed line shows correlation between SWRs and stacked auto-correlograms of all epochs (same as in **g**). **i** Same as **h** but for cell identities shuffled within shanks.

locations) had similar auto-correlograms. This was indeed the case for the three example neurons shown in Fig. 4d. To further test for a possible relationship between individual spike train dynamics and SWR responses, we directly quantified the correlations between the two measures. To this end, principal

component analysis (PCA) was used to reduce the dimensionality of the features (i.e. the number of time bins) of the first 20 ms of auto-correlograms (from wake, REM and NREM epochs) and cross-correlograms with SWRs (±500 ms). The first ten components were considered (explaining 99% and 91% of the variance

of auto-correlograms and SWR cross-correlograms, respectively) (Fig. 4e). The correlation matrix of these weights had visibly more structure than the correlation matrix of PC weights in which neuron identities were shuffled (Fig. 4f; the diagonal blocks of the full correlation matrix are not displayed because PCs are orthogonal to each other). In fact, the total correlation $\rho^2$ of PC weights was significantly higher than shuffles (Fig. 4g; $\rho^2 = 0.31$, $p < 0.001$).

Shuffling neurons recorded simultaneously on a given shank led to a distribution of total correlation that was still significantly smaller than the actual total correlation (Fig. 4g), thus indicating that the relationship between spike train dynamics and responses to SWRs is specific to each neuron rather solely determined by anatomical location. However, the within-shank shuffled correlations were higher than when cell identities were shuffled irrespectively of recording days and shanks ($t = -82$; $p = 10^{-10}$), suggesting some similarity in both auto-correlation and SWR response within each nucleus. The strong correlation at the individual cell level cannot be accounted for by differences in firing rates ($\rho^2_{rate} = 0.3$ vs. $\rho^2 = 0.31$) nor in burstiness alone ($\rho^2_{burst} = 0.3$). Because SWR-associated firing could potentially bias the auto-correlograms during NREM sleep, we repeated the analysis by considering auto-correlograms of only one brain state at a time. The total correlation remained significantly higher than shuffled data ($p < 0.001$) when wake, REM sleep and NREM sleep auto-correlograms were used separately (Fig. 4h-i). Besides, the relationship between auto-correlograms and SWR cross-correlograms holds when HD neurons were excluded ($p < 0.001$, data not shown). These results constitute a demonstration that intrinsic dynamics of individual AT neurons are directly related to their participation in circuit-level activity.

**Spike train dynamics of non-HD AT neurons are heterogeneous**. The link between spike train dynamics and coordination with SWRs, independent of the anatomical location of the neurons (except for HD cells), begs the question of whether there exist different sub-classes of neurons in the AT based solely on the distribution of spike train auto-correlograms. An example HD neuron (cell 1 in Fig. 5a) had a relatively short refractory period across brain states. It also showed a low level of burstiness, if any, during NREM sleep (identified as a peak in the first 8 ms of the auto-correlogram; for the sake of simplicity, burstiness is reported only during NREM sleep), in agreement with previous intracellular studies of AD neurons[32]. In contrast, example cell 2 in Fig. 5a showed a high level of burstiness, as expected for typical thalamic neurons during NREM sleep[25,30], and had a longer refractory period than the example HD neuron (cell 1) during wakefulness and REM sleep. A third cell (cell 2) showed intermediate properties, i.e. a long refractory period but a total lack of burstiness. We thus hypothesized that neurons can be segregated based on their auto-correlograms.

To visually determine the clustering of auto-correlograms, we used t-distributed stochastic neighbor embedding (t-SNE[38],) to project the auto-correlograms from the three different brain states (i.e. three auto-correlograms per neuron) in a two-dimensional embedding. Instead of distinct groups, AT neurons were continuously distributed (along a gradient of burstiness, among other factors) with the exception of HD neurons that formed a separated "island" (Fig. 5b). To confirm this bimodal distribution, we applied the common clustering algorithm, K-means (with $k = 2$), directly to the space of auto-correlograms. The first resulting cluster contained most of the HD neurons (99/127, $p < 0.001$, binomial test). Within cluster #1, the shape of auto-correlograms was highly similar and independent of average firing rates (Fig. 5c). Moreover, the anatomical density of the neurons

belonging to this particular cluster was concentrated around the putative location of the AD nucleus (Fig. 5d). These observations demonstrate that the dynamics of spike train emission, independent of excitability (i.e. firing rate), is sufficient to categorize HD neurons. There was not a direct match between cluster #2 and anatomy, as shown for example by the broad anatomical distribution of average burstiness. These observations were confirmed by the high level of correlation between auto-correlograms of neuronal HD pairs recorded on the same shanks. In contrast, pairs of non-HD neurons from the same shanks had less correlated auto-correlograms (Fig. 5e, $t = 18.32$, $p = 10^{-73}$, t-test).

How much information was sufficient to classify HD versus non-HD neurons? To address this question, we trained automatic classifiers with auto-correlograms to identify the distinction between HD and non-HD neurons. Specifically, we used gradient boosted trees (XGB,[39]), a robust and fast non-linear classifier, on a binary output (HD or non-HD) and we trained classifiers on auto-correlograms of various duration. Classification quality (or "score") was evaluated as a percentage above chance level (0%: chance level; 100%: perfect clustering). HD cells were labeled 50% above chance with only the first 6 ms of spike train auto-correlogram in each of the three brain states (Fig. 5f). This classification on such a limited amount of information suggests that duration of the refractory period and level of burstiness are both unique to HD cells in the AT.

Together, these results provide evidence that HD neurons, in addition to their homogeneous response to hippocampal SWRs and unlike non-HD neurons, share common and specific dynamical properties. This further suggests that AD neurons are a distinct class of neurons in the anterior thalamus while non-HD neurons have broadly distributed properties, irrespective of their nucleus of origin.

**Slow firing dynamics distinguish HD and non-HD neurons**. Although neurons showed a large spectrum of fast timescale dynamics (i.e. faster than 100 ms), does it mean that they were similarly activated during wakefulness on slow ("behavioral") timescales (i.e. on the order of seconds)[40]? These timescales correspond, for example, to the typical firing duration of an HD neuron as an animal rotates its head through the neuron's HD receptive field. For AT HD cells, these timescales are similar during wakefulness and REM sleep[10]. AT neurons showed a wide range of slow dynamics, not only during wakefulness but also during REM sleep, as observed in the various decay times of their auto-correlograms (Fig. 6a). The characteristic timescale, $\tau$, of auto-correlogram decay was well captured by an exponential fit[40]. These behavioral timescales were intrinsic markers of neurons. The average decay times during wake and REM sleep are significantly correlated across neurons (Fig. 6b; $r = 0.434$, $p < 0.001$, Pearson's correlation). HD cells, again, showed marked differences compared with other AT neurons: their intrinsic slow timescales were the slowest (Fig. 6b, inset; $t = -0.23$, $p = 10^{-18}$, t-test for Wake; $t = -0.37$, $p = 10^{-41}$, t-test for REM). In conclusion, neurons of the AT exhibited intrinsic behavioral timescale dynamics, thus possibly reflecting involvement in cognitive functions requiring different integration times[40,41], which is another property that is a sufficient feature to distinguish HD from non-HD neurons.

Spike train dynamics, at fast and slow timescales, must result from complex interactions between intrinsic properties of neurons and network states. It is noteworthy that during wakefulness and REM sleep, thalamic neurons show very rare bursts[25]. Interestingly, burst index (calculated during NREM sleep) was negatively correlated with slow dynamics during both

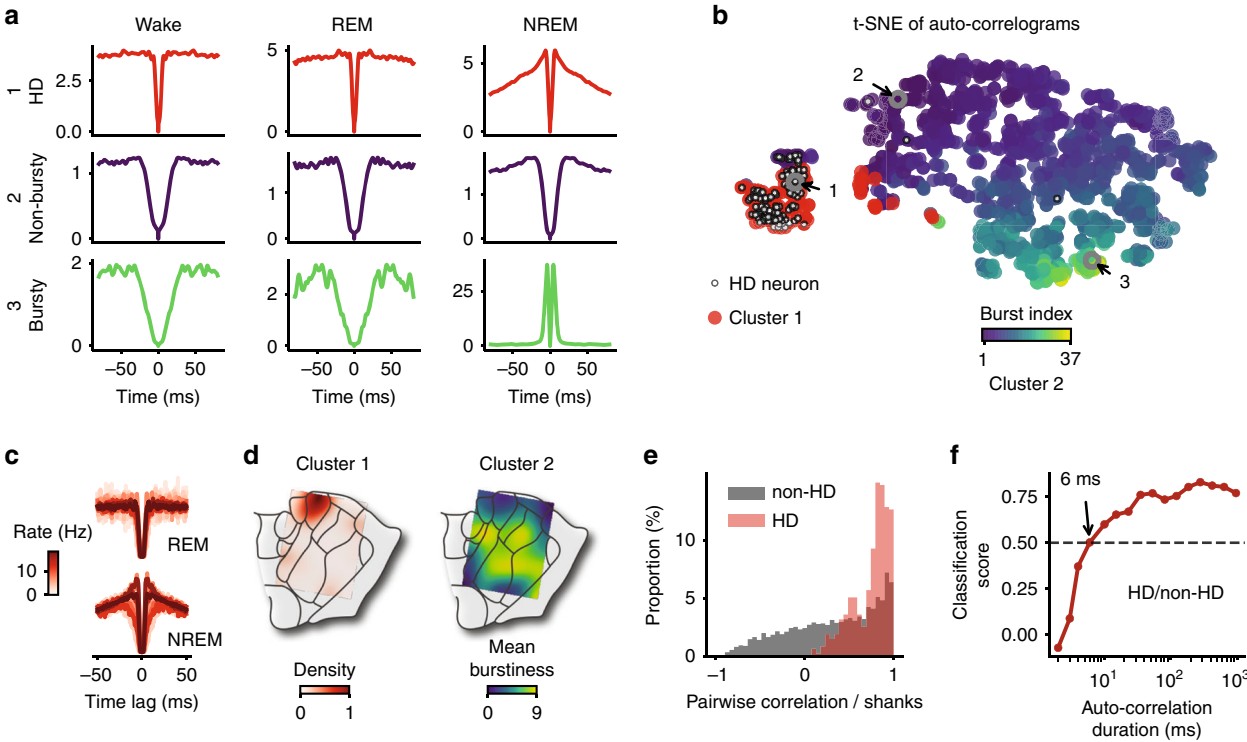

**Fig. 5 Variability and clustering of spike auto-correlograms in the AT. a** Auto-correlograms of an HD (1), non-bursty (2) and bursty (3) neuron during wake, REM sleep and NREM sleep epochs. **b** Clustering and t-SNE embedding of auto-correlograms (each point represents the stacked auto-correlograms from wake, REM sleep and NREM sleep of a neuron). HD neurons are marked with a white dot. The three example neurons from **c** are circled. K-means clustering of the auto-correlograms result in two clusters (see text). Cluster #2 is color-coded relative to burst index during NREM sleep. **c** Left, 15 superimposed auto-correlograms of randomly selected cells from cluster #1 (normalized by the baseline between 50 and 100 ms), during REM (top) and NREM (bottom) sleep. Color (white to dark red) indicates average firing rates (from low to high). Note the high similarity of auto-correlograms during REM sleep, independent of basal firing rates, and the mild variability during NREM sleep. **d** Left, density of cluster #1 (i.e. mostly HD neurons) and, right, mean burst index of cluster #2 over the anatomical schematics, for the same example mouse. **e** Distribution of Pearson's correlation coefficients between pairs of neurons recorded on the same shank showing the dynamical homogeneity of HD neurons compared to non-HD neurons. **f** Classification score of HD versus non-HD neurons (based on auto-correlograms) for increasing duration of auto-correlation. Score is relative to classifiers trained with shuffled data (0, chance level; 1, perfect classification).

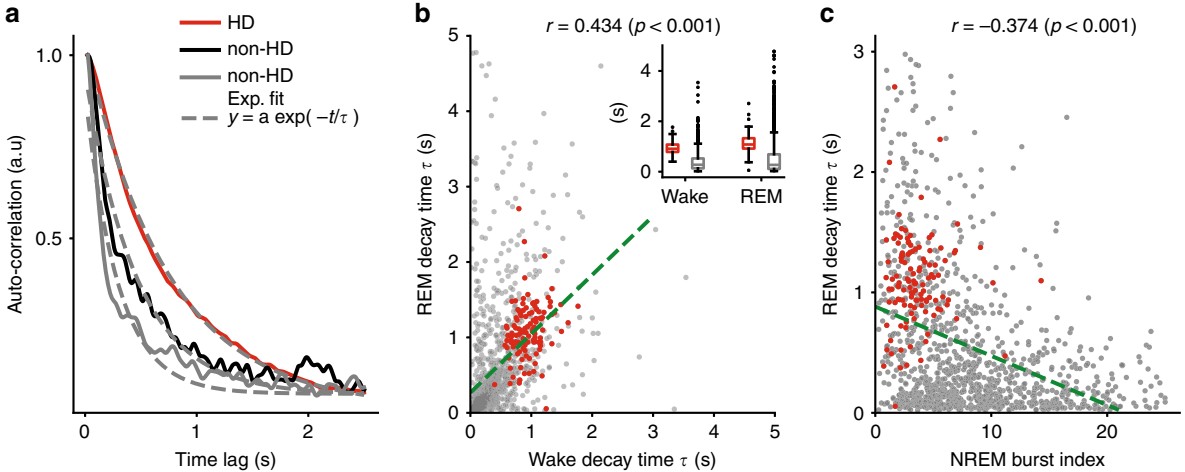

**Fig. 6 Intrinsic slow dynamics during wakefulness and REM sleep in the AT and its relation to fast dynamics during NREM sleep. a** Auto-correlograms of three neurons (HD and non-HD neurons) over long timescale (2.5 s) during wake. Dashed gray lines display exponential fit. **b** Decay time $\tau$ between wake and REM sleep. Red dots indicate HD neurons. Inset shows average (±s.e.m.) decay time for HD and non-HD neurons during wakefulness and REM sleep. **c** Decay time $\tau$ during REM sleep as a function of burst index during NREM sleep ($r = -0.374$, $p < 0.001$, Pearson's correlation). HD cells and non-HD cells are indicated by red and dark dots, respectively.

wakefulness and REM sleep (Fig. 6c during REM sleep; during wakefulness: $r = -0.24$, $p < 0.001$, data not shown). This relationship is potentially due to intrinsic properties which manifest at timescales that are orders of magnitude apart, and suggest that spike train statistics at millisecond timescales recapitulate the firing properties of the neurons in processes taking place at behavioral timescales.

## Discussion

We described how HD cells of the AT (almost exclusively from the AD nucleus) form a homogeneous and specific cell population that plays a key role in limbic processing: while HD cells obviously inform the brain's navigation system of the animal's current HD during wakefulness, they also point towards particular directions at times of hippocampal SWRs during NREM sleep. Thus, they possibly constrain the dynamics of the hippocampus to allow for the generation of coherent replayed trajectories along certain directions. On the contrary, other AT neurons were heterogeneous, even when recorded from the same anatomical location. However, we revealed a link between the dynamics of spike trains and responses to NREM sleep SWRs, thus revealing how intrinsic neuronal properties and circuit integration are certainly related in the AT. One limitation of our study is that we only analyzed NREM sleep SWRs. The behavioral task was not designed to maximize the number of awake SWRs, and further studies are needed to determine the relationship between HD and non-HD neurons of the AT and awake SWRs.

Here, we used a non-linear dimensionality reduction technique, ISOMAP[42], to decode population states independent of their behavioral correlates[26]. We showed how the HD cell population of the AD nucleus, which normally drifts at high angular speed during NREM sleep[10,26], stabilized at fixed points and at high gain immediately before SWRs. It remains to be shown whether HD cells firing in the AD can in fact orchestrate hippocampal activity. However, this transient gain increase in the HD cell population preceding SWRs was independent of the relative timing within the UP states. This suggests that the AD nucleus can rapidly influence hippocampal dynamics, and not only at the timescale of the slow oscillation[21,23,24,34]. While the impact of AD firing on the timing of SWRs is indirect and through polysynaptic routes, the projection fields of AD neurons, including targets in the post-subiculum, retrosplenial cortex and medial entorhinal cortex, suggest a broad contribution of AD neurons to the activity of the limbic system, as evidenced by their central role in synchronizing the slow oscillation during NREM sleep[12].

Not only do AD neurons exert a widespread influence on the limbic system during NREM sleep, they also convey a coherent message. Indeed, the population activity of HD neurons of the AD nucleus is still highly organized during NREM sleep[10,26] in such a way that a virtual HD can be reliably decoded at any time with the exception of DOWN states that correspond to a singularity in HD cell population dynamics. It is thus possible that spatially tuned neurons of the limbic system are also updated by the HD signal during NREM sleep. While the HD cell population shows fast "sweeps" during NREM sleep[10,26], the stabilization of the HD signal during SWRs suggests that the navigation system is transiently constrained in a particular direction. While our study did not include neuronal recordings in downstream structures, one prediction is that replay events of the hippocampus[15,16] correspond to linear bouts of possible trajectories. Following exploration of a two-dimensional environment, it was recently suggested that replayed trajectories follow random (and non necessarily linear) rather than previously experienced paths[16]. Whether the directions of the replayed trajectories, from start to end points, or the initial direction of the replayed trajectories correspond to the direction encoded by the HD network remains an open question. Another possibility is that the HD cells influence replay in the medial entorhinal cortex. Indeed, grid cells maintain their coordination during NREM sleep[43,44] and replay previously formed trajectories[45,46], possibly independent of the hippocampus[46]. Finally, the coherent HD signal provided by the AD nucleus may constitute a subcortical process of coordinated neuronal sequences during sleep that do not correspond to any experienced spatial trajectories[16,47]. Paired recordings of AT HD neurons and place or grid cells will be necessary to test these predictions.

One outstanding question is whether or not the subcortical HD network itself is under the influence of an external signal related to SWR generation and content. Indeed, one view posits that the HD network, including the AD nucleus and upstream structures, randomly fluctuates during sleep by integrating noisy inputs[26]. A transient fluctuation in excitability would favor the occurrence of a SWR through polysynaptic pathways. Conversely, it is possible that such a state is controlled by cortical feedback: AD neurons, as well as their presynaptic neurons of the lateral mammillary nuclei, receive a feedback from the post-subiculum. However, post-subicular cells remain largely under the influence of the AD nucleus during NREM sleep[27]. At any rate, AD alone is certainly not the only structure influencing SWRs. For example, auditory cortex neurons fire prior to SWR and predict the content of hippocampal replays[48].

While the exact input and output connectivity pattern of each AT neuron is intractable in vivo, their coordination with hippocampal activity is an opportunity to determine their functional integration within the circuit. Unlike HD neurons, the co-modulation with SWRs was broadly distributed for non-HD neurons, largely independent of their anatomical location. One prediction is that with the exception of the AD nucleus, other cytoarchitecturally defined AT nuclei are characterized by a large diversity of connectivity patterns from neurons to neurons, as previous anatomical studies have suggested[1,29].

The flow of information in the limbic system, and in particular to and from the hippocampus, may vary across brain states, especially between wakefulness and NREM sleep[49]. AT neurons showed similar responses to SWRs (during NREM sleep) and theta (during wakefulness and REM), thus suggesting that they play a role in routing information independently of brain states. Whether this results from specific coupling with the hippocampus or from modulation by other limbic structures (e.g. the medial septum) remains to be answered.

Surprisingly, spike train dynamics are another potential marker of the functional coupling of AT neurons with the hippocampus. In fact, we observed a strong relationship between spike train auto-correlograms across brain states and responses to SWRs. While our study could not determine the intracellular properties of AT neurons, spike train dynamics directly reflect – at least in part – these intrinsic properties[50,51]. Again, HD cells formed a highly homogeneous population of neurons that could be identified as separate from the non-HD cells. In contrast, non-HD cells showed a continuous spectrum of dynamics with graded burstiness and refractory periods, among other features. Furthermore, AT neurons show a wide range of slow (~1 s) intrinsic timescale dynamics. The intrinsic nature of these timescales is suggested by the similar time constant of their spike train auto-correlogram during wakefulness and REM sleep, two activated states of the brain. HD cells show markedly longer behavioral timescales than other AT neurons. Together, these results suggest that the cytoarchitectural definition of AT nuclei is an insufficient level of description with the exception of the AD nucleus in which neurons show homogeneous behavioral correlates, for functional coupling with the hippocampus and spike train dynamics. In

support of this view, gene expression profile in the AD is unique in the thalamus, while other AT neurons are continuously distributed rather than clustered in cytoarchitecturally defined nuclei[52].

A growing body of evidence points towards an intimate link between gene expression, spontaneous activity and circuit development in thalamocortical pathways[29,52–55]. It is thus possible that spike train dynamics and coupling to network activity are two facets of common processes that are directly related to the acquired (but not necessarily pre-configured) role that each neuron plays in the circuit and associated with a specific molecular makeup[29,52–57]. Whether the variability in spike train dynamics reflects specific processes for signal transmission to downstream readers remains an open question.

The AT is a key communication relay between structures of the limbic system[7]. Interactions between the thalamus, the hippocampus and more generally the limbic system during learning and around SWRs (nested in slow and spindle oscillations, two NREM-specific thalamocortical rhythms) are believed to be crucial for supporting hippocampus-dependent memory consolidation processes at play during NREM sleep[12,13,17,21,22,24,28,58,59]. Interestingly, while the HD cell population is stabilized at times of SWR occurence, it drifts at maximum speed during thalamocortical spindles[26]. Future work is necessary to characterize the processes controlling HD and non-HD cells of the AT, their contribution in determining hippocampal replay content and their role in routing hippocampal signals across the structures of the limbic system. The analytical framework of our study provides a foundation for investigating the relationship between intrinsic neuronal properties at various timescales and functional integration in brain networks in vivo. It is possible that such a relationship is an organizational principle of all thalamic neurons.

## Methods

**Surgery and experimental design**. All experiments were approved by the Institutional Animal Care and Use Committee of New York University Medical Center. Details of the surgeries were described previously[10]. Briefly, four male mice weighing ~30 g (3–6 months) were implanted under isoflurane anesthesia with silicon probes (Neuronexus) above the anterior thalamus (AP: −0.6 mm; ML: −0.5 to −1.9 mm; DV: 2.2 mm, with a 10–15° angle, the shanks pointing toward midline). Hippocampal wire bundles were implanted above CA1 (AP: −2.2 mm; −1 to −1.6 mm ML; 1 mm DV). The probes consisted of 8 shanks separated by 200-µm and each shank had eight recording sites (160 µm² each site, 1–3-MΩ impedance) that were staggered to provide a two-dimensional arrangement (20-µm vertical separation). In all experiments, ground and reference screws or 100-µm diameter tungsten wires were implanted in the bone above the cerebellum.

**Electrophysiological data acquisition**. The animals were recorded over several days for successive epochs of pre-sleep (1–2 h), food foraging in a circular arena (sweetened cereals or regular food pellets) for 30 min, and post-sleep (1–2 h). Overall, sessions lasted on average 5 h (±0.94 s.d.). Electrophysiological signals were acquired continuously at 20 kHz on a 256-channel Amplipex system (Szeged; 16-bit resolution, analog multiplexing). The broadband signal was down sampled to 1.25 kHz and used as LFP.

To track the position of the animals in the open maze and in their home cage during rest epochs, two small light-emitting diodes (LEDs; 5-cm separation), mounted above the headstage, were recorded by a digital video camera at 30 frames per second. The LED locations were detected online and resampled at 39 Hz by the acquisition system. Spike sorting was performed semi-automatically, using KlustaKwik (http://klustakwik.sourceforge.net/). This was followed by manual adjustment of the waveform clusters using the Klusters software.

**Sleep scoring**. As previously described[10], stages of sleep were identified semi-automatically based on the CA1 LFP spectrogram and on animal movements that were continuously tracked with the LEDs, as during behavior. Overall, sleep was defined as a long period of immobility. Within each sleep episode, NREM sleep was defined as periods with high delta (1–4 Hz) and spindle (10–15 Hz) activity. REM sleep was defined as periods with strong power in the theta (5–12 Hz) range and low delta.

**UP and DOWN state detection**. UP and DOWN states were detected by computing the total firing rate of all simultaneously recorded neurons in bins of 10 ms, smoothed with a Gaussian kernel of 20 ms s.d. Epochs during which total firing rate was lower than 20% of the maximum firing rate were considered as DOWN epochs. Epochs shorter than 30 ms and longer than 500 ms were discarded. UP states were defined as the epochs between each DOWN states.

**SWR detection**. To detect SWRs, CA1 LFP was first bandpassed between 80 and 300 Hz with a Gaussian filter. The squared signal was then smoothed using a digital filter with a window length of 11 and z-scored using mean and standard deviation. The normalized squared signal was then thresholded between 3 and 7 standard deviations yielding a first set of candidate ripples. The set was then reduced by keeping only candidates with a duration between 25 ms and 350 ms (in some sessions with lower amplitude ripples, the thresholds were lowered to 2 and 5 s.d., respectively, and rippled detection was validated by visual inspection). Ripples closer than 30ms were merged and considered a unitary event.

**Head-direction classification**. For each session, the direction of the head of the animal in the horizontal plane was calculated by the relative orientation of a blue and red LED located on top of the head. Head-direction neurons were detected by computing tuning curves i.e. the ratio between the histogram of angular direction associated to each spike divided by the total time spent by the animal in each angular bins. Similar to Peyrache et al.[10], a Rayleigh test was performed to test for the null hypothesis of uniformly distributed firing in all angular directions, and neurons were classified as HD cells if peak firing rates of the tuning curves was >1, a probability of non-uniform distribution <0.001 and a concentration parameter (i.e. inverse of the variance of the tuning curves) larger than 1.

**ISOMAP projection and decoding**. ISOMAP projections (Fig. 1 and Supplementary Fig. 1) were performed for sessions containing at least 10 HD neurons (eight sessions in total from three different animals). First, spike trains were binned during wakefulness and around time of SWRs. As such dimensionality reduction techniques are computationally heavy, we included in the analysis only the first 15 min of exploration and a period of 1 s around SWRS (±500 ms). The duration of the bins was 400 ms and 30 ms (with a 50% overlap) during wakefulness and around SWR occurrence times, respectively (different bins were used to capture the different timescales of the HD signal during wakefulness and NREM sleep). To visually inspect the topology of HD cell population vectors (Fig. 1c and Supplementary Fig. 1a), we used a bin size of 200 ms for wakefulness and 100 ms (with an overlap of 75%) around SWRs. To quantify the change in radii and angular velocity across sessions, ISOMAP projections were compared to a baseline taken around randomly selected events during NREM sleep. Similar to Chaudhuri et al.[26], we computed the square root of the rates to normalize for the variance in firing rates. Binned firing rates were smoothed with a Gaussian kernel of three bin standard deviation (independent of absolute bin duration).

In each condition (wakefulness, SWR, and controls), binned firing rates were stacked together, yielding a rate matrix $R \in \mathbb{R}^{T \times N}$, where $N$ is the number of neurons and $T$ is the total number of time points. This rate matrix was then projected to a two dimension plane $I \in \mathbb{R}^{T \times 2}$ using ISOMAP[42]. The number of neighbors was set to 200. To further reduce the computational requirement for long spike trains (i.e. typically more than 30000 time bins), we repeated the ISOMAP projection using a random subset of 150 SWRs and 150 random NREM sleep time bins for baseline quantification with the same 15 min of awake data, until all SWRs were analyzed. This ensures that at each iteration, ISOMAP gives a similar topology of embeddings and that neuronal "trajectories" around SWRs can be compared to each other for a given session.

Angular direction and radius (i.e. ring size) was decoded during SWRs by computing the element-wise arc tangent and Euclidean norm of each time point. Angular velocity was then evaluated by computing the angular difference between two consecutive time points. The same procedure was performed for randomly selected events during NREM sleep that projected within the same ISOMAP embedding. Radius and angular velocity (as shown in Fig. 1d, e) are thus the ratio $(\bar{X}_{\text{SWRs}} - \bar{X}_{\text{random}})/\bar{X}_{\text{random}}$ with $\bar{X}$ denoting average values).

**Bayesian decoding**. To validate the decoding in the embedding space (Fig. 1), we used Bayesian decoding to predict angular velocity during SWRs (Fig. 2). Let $\mathbf{n} = (n_1, n_2, \ldots, n_N)$ be the number of spikes fired by the HD neurons within a given time window (30 ms) and $\Phi$ be the animal's head direction or, during sleep, the angle of the internal HD signal. The goal of this decoder is to compute the posterior probability $P(\Phi|\mathbf{n})$, which can be achieved using Bayes' rule of conditional probabilities:

$$P(\Phi|\mathbf{n}) = \frac{P(\mathbf{n}|\Phi)P(\Phi)}{P(\mathbf{n})} \tag{1}$$

Assuming that (i) neuronal firing is independent from each other and (ii) spike counts follow Poisson distributions, the probability $P(\mathbf{n}|\Phi)$ is equal to[33]:

$$P(\mathbf{n}|\Phi) = \prod_{i=1}^{N} P(n_i|\Phi) = \prod_{i=1}^{N} \frac{(\tau f_i(\Phi))^{n_i}}{n_i!} \exp^{-\tau f_i(\Phi)} \tag{2}$$

where $\tau$ is the bin size and $f_i(\Phi)$ the average firing rate of cell $i$ for the direction $\Phi$ during wakefulness (i.e. the HD tuning curve at angle $\Phi$). The decoded direction is the value $\Phi$ associated with the highest posterior probability $P(\Phi|\mathbf{n})$.

Similar to ISOMAP decoding, angular velocity was evaluated as the ratio $(\bar{v}_{SWRs} - \bar{v}_{random})/\bar{v}_{random}$ with $\bar{v}_{SWRs}$ being the average angular velocity during SWRs and $\bar{v}_{random}$, the average angular velocity for randomly selected events during non-REM sleep. We used bin size of 30 ms with 50% overlap and angular bins of 6 degrees.

**Map alignment**. To align the position of the electrodes with their putative anatomical location, we used the following procedure. First, after each session, the probe was lowered by 70–140 µm. Relative position was estimated from the first day of recording when the probe entered the thalamus (appearance of spiking activity on 1–2 shanks). This yielded a relative depth for each session while the horizontal distance between shanks was kept constant (200 µm). This two-dimensional grid of recording sites was then rotated by 15° clockwise corresponding to the angle of penetration of the silicon probe. Then, the relative position of recording sites was aligned to the anatomical map (from bregma −0.82 mm; Fig. 38 in Paxinos et al.[60]) by matching the relative configuration of the electrodes with the putative location of the AD nucleus (shanks on which large number of head-direction neurons were recorded)[10]. Note that the anatomical map from the atlas was enlarged by 10% (it is commonly accepted that slices used for anatomy are shrunk by approximately this amount). As shown in Figs. 2 and S2, the density of HD neurons matched well with the estimated anatomical position of the anterodorsal nucleus.

**SWRs cross-correlograms**. The SWR modulation was computed for each neuron by estimating cross-correlograms (average firing rates in 5 ms bins, ±500 ms from SWR peak time). Because neuronal discharge and SWRs are both co-modulated by the slow oscillation[21–24], cross-correlograms were normalized relative to their expected values under the null hypothesis of no short timescale coupling between neuronal discharge and SWR occurrence. To this end, SWR cross-correlograms were convolved with Gaussian windows of 150 ms s.d. (a process similar to low-pass filtering the firing rates). The value of 150 ms was chosen as it corresponds to the upper bound duration of SWRs. From this distribution of low-pass filtered ("expected") rates around the time of SWRs, we inferred an "expected", standard deviation under the assumption of a Poisson process (i.e. the square root of the expected rate at a given time bin). For each time bin, we subtracted the expected rate from the observed rate, and then divided the difference by the expected standard deviation. The observed cross-correlograms were thus expressed in $z$-values from the expected distribution under the null hypothesis of no short timescale coupling. This method enables us to extract the specific and fast modulation of a neuron by SWRs, independent of the co-modulation of the SWRs with the thalamocortical slow oscillations[22–25]. In mathematical terms, we thus obtained a set of time series $\mathbf{z}(t) = [z_1(t), \ldots, z_N(t)]$ where $N$ is the number of neurons. Note that the $z$ values are not a typical z-score, but correspond to deviation in z from the null hypothesis. Note that the mean and s.d. of the resulting z-transforms are not normalized. To describe the amplitude of SWR modulation, we introduced SWR energy, defined as the total power of the z-scored cross-correlograms:

$$|z_i| = \sqrt{\sum_{t=1}^{T} z_i(t)^2}.$$

**Theta modulation**. Theta modulation for each neuron was computed separately for epochs of wake and REM sleep. Using continuous wavelets transform, a phase was assigned to each theta cycle and histograms of spike counts between 0 and $2\pi$ were then computed for each neuron as shown in Fig. 3b. A Rayleigh test was then performed with the null hypothesis of uniformly distributed firing rates during a theta cycle.

**jPCA**. To describe SWR modulation in term of phases, we used the jPCA method[37]. First, PCA was used to reduce the dimensionality of the ensembles of cross-correlograms ($K = 6$ principal components were considered, as in Churchland et al.[37]). Overall, the goal of jPCA is to describe temporal response profiles, assuming that the data are governed by a linear dynamical system of the form: $\dot{\mathbf{x}} = M\mathbf{x}$ with $M \in \mathbb{R}^{K,K}$ (or $M \in \mathbb{R}^{N,N}$ if no dimension reduction is applied first). The matrix M can be split into its symmetric and anti-symmetric matrix $M = M_{sym} + M_{anti}$, related to expansion/contraction and rotational dynamics, respectively. The latter matrix has pure imaginary eigenvalues, hence its association with rotational dynamics. To capture phase response in the rotational space, it is thus sufficient to find the best fit for the matrix $M_{anti} \in \Gamma^{k \times k}$. This can be done by minimizing the error made on predicting $\dot{X}$ from $MX$, which can be expressed as $M_{anti}^* = \mathrm{argmin}_{M \in \Gamma}||\dot{X} - MX||_F$, where $F$ denotes the Froebenius norm (see Churchland et al.[37] for additional details on the methods). Finally, jPCA entails the decomposition of $M_{anti}^*$ into its eigenvectors made (by definition) of complex conjugate pairs. A suitable pair of projection vectors $U = (u_1, u_2) \in \mathbb{R}^{T,2}$ can be obtained by combining complex conjugate pairs $(v_1, v_2)$ such that $u_1 = v_1 + v_2$ and $u_2 = i(v_1 - v_2)$.

The last step consists in projecting SWR cross-correlograms onto the first two jPCs ($u_1$ and $u_2$): $\mathbf{y} = Z^T\mathbf{u} \in \mathbb{R}^{N,2}$. The SWR phase was defined as the angle from origin in this 2D space: $\phi_{SWR} = \mathrm{atan2}(y_2, y_1)$.

**Auto-correlograms**. For each neuron, three auto-correlograms were computed separately for the epoch of wake, REM and NREM sleep respectively. The short auto-correlograms shown in Figs. 4 and 5 were computed using a bin size of 0.5 ms while the long auto-correlograms in Fig. 6 were computed using a bin size of 5 ms.

Embedding of the auto-correlograms in a 2D map (Fig. 5b) was performed with the t-SNE algorithm[38] by concatenating only the part corresponding to positive time lag (i.e. the right part of the auto-correlograms). Given $A_{2\,ms\to40\,ms}^{epoch} \in \mathbb{R}^N$ the auto-correlogram vector of one epoch for one neuron, each point in the t-SNE projection is thus a mapping $f : X_i \in \mathbb{R}^{3N} \to Y_i \in \mathbb{R}^2$ with $X_i = [A_{2\,ms\to40\,ms}^{wake} A_{2\,ms\to40\,ms}^{REM} A_{2\,ms\to40\,ms}^{NREM}]$ for each neuron i.

In Fig. 5f, neurons were classified (HD versus non-HD) with gradient boosted trees using the XGBoost package[39]. Inputs given to the classifier is the same as the t-SNE algorithm (i.e. stacked auto-correlograms of wake, REM and NREM sleep episodes). Classifier was trained with 1000 iterations, using default parameters and a 10-fold cross-validation procedure. Chance levels were determined from a thousand classifications obtained with random shuffling of the neuron labels. For N neurons and C classes, classification score was defined as:

$$s = \frac{\sum_{i=1}^{N} 1(c_i = \hat{c}_i) - \sum_{i=1}^{N} 1(c_i = \hat{c}_i^s)}{N - \sum_{i=1}^{N} 1(c_i = \hat{c}_i^s)}$$

with $1(x)$ the indicator function, $\hat{c}_i$ the class predicted for the $i$th neuron and $\hat{c}_i^s$ the labels predicted with a classifier trained on shuffled data. A score of 0 indicates chance level (compared to shuffles) and a score of 1 indicates perfect classification.

**Burst index**. For each epoch, bursts were defined as groups of spikes with interspike intervals (ISI) between 2 ms and 9 ms. Burst index[61] was then computed as the ratio between the observed count $N_{2\,ms<ISI<9\,ms}$ and the count expected from a homogeneous process with the same average firing rate: $\bar{N}_{\tau_1<ISI<\tau_2} = T(exp(-\tau_1 \times r) - exp(-\tau_2 \times r))$ with $T$ the duration of the epoch, $r$ the mean firing rate during this epoch, $\tau_1 = 2$ ms and $\tau_1 = 9$ ms.

**Auto-correlograms and SWR cross-correlograms correlation**. To compute the correlation between SWR cross-correlograms and auto-correlograms (Figs. 4e–g), auto-correlograms were first stacked across brain states (as done for the t-SNE projection, see above). We used PCA to reduce the dimensionality of each matrix (of either SWRs cross-correlograms or stacked auto-correlograms) and we kept the first 10 components accounting for respectively 91% and 99% of the variance. We computed the 20-by-20 correlation matrix $C$ for the 10 SWR PCs and 10 auto-correlogram PCs. The matrix is diagonal for the upper-left and bottom right 10-by-10 blocks as PCs from a given dataset are perpendicular to each other. We thus show only the off-diagonal matrix in Fig. 4f. We then defined the total correlation as $\rho_{total}^2 = 1 - |C|$ with $|C|$ being the determinant of the correlation matrix (if only one PC was considered for auto-correlograms and SWR cross-correlograms, then total correlation would be the linear correlation between the two PCs). From a geometrical perspective, this measure captures the fraction of state-space volume occupied by both measures. The null correlation was determined by shuffling the identity of the neurons in both datasets (1000 times).

We controlled for correlation between neurons belonging to the same nucleus by shuffling neurons within groups recorded on the same shank and session (1000 times). The correlation between SWR cross-correlograms and auto-correlograms was further tested for possible correlation by common underlying factors (firing rates or burstiness). The correlation $r_{ij}$ between the $i$th auto-correlogram PC and the $j$th SWR PC was replaced by the partial correlation (correlation knowing factor $f$): $r_{ij|f} = \frac{r_{ij} - r_{if}r_{jf}}{\sqrt{1 - r_{if}^2}\sqrt{1 - r_{jf}^2}}$ where $r_{kf}$ ($k = i$ or $j$) is the correlation between PC weights and the external factor $f$ (firing rates or burstiness).

**Quantification and statistical analysis**. Analyses were done using customized code written in MATLAB (MathWorks) and Python (3.5) with the following libraries: numpy, scipy and scikit-learn.

**Reporting summary**. Further information on research design is available in the Nature Research Reporting Summary linked to this article.

## Data availability
The data that support the findings of this study are available on http://crcns.org/data-sets/thalamus/th-1 (https://doi.org/10.6080/K0G15XS1). We used animals labeled Mouse17, Mouse12, Mouse20 and Mouse32 in the dataset. Code is available at https://github.com/PeyracheLab/ThalamusPhysio.

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

## Acknowledgements

The authors thank G. Buzsaki, A. Adamantidis, F. P. Battaglia, M. D. Humphries, L. Sjulson, R. Chaudhuri, D. E. English and J.-P. Roussarie for critical reading of the manuscript, P. C. Petersen and M. M. Lacroix for help with the experiments, members of the Peyrache Lab for comments and for editing the manuscript. A.P. holds a Canadian Research Chair in Systems Neuroscience. This work was supported by CIHR Project

Grant 155957, NSERC Discovery Grant RGPIN-2018-04600 and IRDC Project Grant 108877-001.

## Author contributions

A.P. conceived and designed the study. A.P. performed the experiments. G.V. and A.P. analyzed the data. G.V. and A.P. wrote the manuscript.

## Competing interests

The authors declare no competing interests.
