## [Peer Review File · Nature Communications]

Reviewers' Comments:

Reviewer #1:

Remarks to the Author:

This work revealed unexpected coordination of the anterior thalamus (AT) head direction system with hippocampal sharp-wave ripples (SWR). They showed that prior to SWR onset in the hippocampus a selected subset of AT head direction cells increased their activity which all encode a similar head direction. Hence prior to hippocampal SWR-associated reactivation the AT signals a coherent head direction, which might also guide which trajectories are reactivated in the hippocampus. They also show that unlike head direction cells, non-head direction AT cells exhibited variable SWR response profiles and spike timing dynamics. Interestingly, just using the autocorrelations of cells head direction cells could be differentiated from other cell types. Moreover, AT cells with similar autocorrelations show similar SWR responses. This work suggests a surprising and unexpected subcortical AT influence to hippocampal reactivation and hence point to complex brain-wide interactions during SWR associated memory consolidation. I have only one major comment and few minor suggestions.

Major comment

Although I understood the PCA analysis in Figure 4 e-h, this and the underlying rationale could be better explained in the results. I also wonder to what degree this result is due to the coherent autocorrelations and SWR response profile of the head direction cells only. Most likely it is not the case, but this analysis could be performed for non-head direction cells only to see whether the finding stands for these cells only.

Minor

Introduction 5th paragraph –spell out that the ‘coherent head direction signal’ is not related to the current head direction of the animal.

Page 3 last paragraph ‘radii’ is possibly a typo.

Figure 2e please use the same y scale for both figures to allow proper comparison

Figure 3 more explanation is needed for the jPCA analysis Also, it is unclear how to interpret this result.

Page 7 4th paragraph – I assume theta refers to REM sleep here

Reviewer #2:

Remarks to the Author:

In the present manuscript authors report that head-direction (HD) neurons in the anterior thalamus (AT) present a homogeneous subpopulation of AT neurons that can be distinguished from other AT neurons by their intrinsic properties and coupling with population activity in the hippocampus. Specifically, HD cells were clustered in the anterior dorsal (AD) thalamus, exhibited distinctive spike train dynamics (namely, short refractory period and lack of burst firing), their activity was not modulated by hippocampal theta oscillations, but it was tightly coupled with hippocampal sharp-wave ripples (SWRs). About 8% of all recorded neurons in AT were classified as HD cells; the remaining neurons (> 90% of recorded neurons) were assigned to non-HD cells. The non-HD cells presented a heterogeneous population with broadly distributed properties. Their activity was differentially modulated around SWRs; a large proportion of non-HD cells (38%) was phase locked to hippocampal theta oscillations. The difference in spike train dynamics between HD and non-HD cells was also revealed at a slow time scale. Due to specific coupling of HD cells with the hippocampal population activity as revealed by SWRs, authors suggest that this AT cell population plays a key role in limbic offline processing by shaping ensemble replay. Overall, it is an interesting study, the results are novel, and the manuscript is well written. The data were analyzed using conventional, but also a number of refined algorithms. Most importantly, authors describe intrinsic properties of a relatively large population of AT neurons (> 2000) and report substantial diversity between individual AT neurons even within the same anatomically defined nucleus. In addition, authors linked the intrinsic properties of AT neurons with their coupling with hippocampal population activity (theta and SWRs). Large-scale thalamic recordings in behaving animals remain rare; therefore, this study is definitely a big and valuable achievement and will present interest to a broader audience.

Major concerns:

- Throughout the text, it is not always clear if authors considered awake or sleep ripples or both (except when Up and Down states were identified). Was thalamic modulation different around awake or sleep ripples? Did modulation depend on ripple features as amplitude?

- Figure 1b shows that at time of a given SWR some HD cells were active while others were not. Figure 2d shows that overall all HD cells ($n = 161$) appear to be SWR-modulated. How consistent was modulation of a given HD cell across successive SWRs? If I understood correctly, authors claim that at time of a given SWR a subset of HD cells with similar orientation would be co-activated. If this is correct, I suppose, authors do not have sufficient amount of simultaneously

recorded HD cells to illustrate this point. If such coherent activation of HD cells with the same orientation exists offline, is it hard-wired? This point shall be further clarified and maybe addressed in discussion.

- Figure 2d shows that a proportion of non-HD cells were also SWR-modulated? Plotting the distribution of SWR 'energy' for non-HD vs HD cells would be helpful to assess the modulation strength across these two thalamic populations.
- Have authors attempted to cluster the modulation profiles of non-HD cells? One can suspect a more complex temporal dynamics of modulation profiles of different populations of AT (non-HD) neurons, as seen on Figure 2d. In general, I find averaging across all non-HD cells (> 90% of recorded cells) not justified, if not misleading.
- Concerning theta modulation: Do AT neurons (page 7) include HD cells? Was correlation between SWR energy and theta modulation significant when HD cells only were considered?
- Authors' statements about causality of cross-regional interactions are not supposed by presented results and need to be toned down. For example, page 6, last paragraph and also in the discussion.

Minor

- Abstract: I suggest to revise the sentence to make it more informative "...HD cells ... are set in stable states ... before SWRs". Authors need to clarify what they mean by "stable states". Same in the last paragraph of Introduction.
- NREM is not always followed by sleep to read NREM sleep.
- Page 2, line 5: revise "...HD cells remain coordinated during sleep..." as it is not clear what authors mean without consulting referenced papers. Same in the second paragraph of Results: "... convey a coherent signal..." phrase not informative without reading referenced papers. Please, revise to make its meaning clearer.
- Figure 2c: why number of SWRs is limited to 100. Are they randomly selected or successive SWRs?

We would like first to thank the referees for their constructive comments. Please find below our point- by-point response in blue characters.

Reviewers' comments:

Reviewer #1 (Remarks to the Author):

This work revealed unexpected coordination of the anterior thalamus (AT) head direction system with hippocampal sharp-wave ripples (SWR). They showed that prior to SWR onset in the hippocampus a selected subset of AT head direction cells increased their activity which all encode a similar head direction. Hence prior to hippocampal SWR-associated reactivation the AT signals a coherent head direction, which might also guide which trajectories are reactivated in the hippocampus. They also show that unlike head direction cells, non-head direction AT cells exhibited variable SWR response profiles and spike timing dynamics. Interestingly, just using the autocorrelations of cells head direction cells could be differentiated from other cell types. Moreover, AT cells with similar autocorrelations show similar SWR responses. This work suggests a surprising and unexpected subcortical AT influence to hippocampal reactivation and hence point to complex brain-wide interactions during SWR associated memory consolidation. I have only one major comment and few minor suggestions.

Major comment

Although I understood the PCA analysis in Figure 4 e-h, this and the underlying rationale could be better explained in the results. I also wonder to what degree this result is due to the coherent autocorrelations and SWR response profile of the head direction cells only. Most likely it is not the case, but this analysis could be performed for non-head direction cells only to see whether the finding stands for these cells only.

Interesting point. The result holds after excluding HD neurons (see figure on the left). The vertical red line shows the true covariation between single-cell spike train auto-correlograms and SWR cross-correlograms for all units except HD neurons. The black distribution shows the covariance after random shuffling of cell identities.

We have added the following sentence in the main text:

"The relationship between auto-correlograms and SWR cross-correlograms holds when HD neurons were excluded ($p < 0.001$, data not shown)."

Minor

Introduction 5th paragraph –spell out that the ‘coherent head direction signal’ is not related to the current head direction of the animal.

We appreciate the referee’s suggestion. We have updated the sentence to clarify this point:

“This suggests that during sleep AD neurons continue conveying a coherent HD signal to the navigation system, including the hippocampus, independent of the current heading of the animal.”

Page 3 last paragraph ‘radii’ is possibly a typo.

The referee is correct. We have changed the sentence to:

“Average radius was maximal before SWR peaks (-17ms +/- 6 ms; t=-2.82, p=0.03; t-test).”

Figure 2e please use the same y scale for both figures to allow proper comparison

The y scale has been updated in figure 2e.

Figure 3 more explanation is needed for the jPCA analysis Also, it is unclear how to interpret this result.

The jPCA analysis was first developed by Churchland et al. (2012) to capture the rotational trajectory of a dynamical system. They applied this method to neural activity during a non-periodic behavior (i.e. a reaching task). For different arm movements, the authors observed different types of rotational trajectory. Thus, they suggested that the jPCA projection captures the various polyphasic responses of a population of neurons in a simple two-dimensional trajectory, describing the temporal evolution of spiking activity during a single “pseudo-cycle”.

In our study, we aimed at comparing the neural activity of the thalamic population during SWR, a non-periodic event, and theta oscillations, which repeat regularly cycles after cycles. The modulation of a population of neurons relative to an oscillation is usually described in terms of preferred phase and amplitude of spiking modulation. We hypothesized that this temporal dynamics (i.e. the order and magnitude of firing activity) is the same during SWRs than during a theta cycle. However, a SWR event is not an oscillation and a preferred phase cannot be directly measure. To reveal the underlying temporal dynamics of a population of neurons around SWRs, we projected their cross-correlograms onto a two-dimensional jPCA space allowing us to determine a pseudo preferred phase. Neurons with similar SWR cross-correlograms were clustered together in the jPCA space and assigned a similar phase (see Fig. 3e and Fig. 3g).

Interestingly, SWR jPCA and theta phases are correlated (Fig. 3h). We interpreted this result as follow: Although theta oscillations and SWRs occur in different brain states, the anterior thalamic neurons respond (or participate) in a similar way to a single theta cycle and a SWR. This suggests that the relationship between the anterior thalamus and the hippocampus is brain-state independent. Furthermore, as the thalamus lacks the kind of local dynamics that exist in the cortex (no excitatory collaterals, single type of inhibitory cell type, etc.) these invariant responses possibly reflect the connectivity of the circuit. It is tempting to assume that neurons firing at different phases

are preferentially connected (possibly via polysynaptic routes) to different parts of the hippocampus and entorhinal cortex (e.g. dorsal versus ventral), but this remains speculative for now.

We have rephrased the corresponding paragraph in the main text to try to justify more clearly this choice of analysis:

Firing of AT neurons relative to SWRs cannot, overall, be trivially described (e.g. excited or inhibited). Rather, they show a wide range of temporal profiles (Fig. 3e, for theta-modulated neurons). While modulation by oscillations is commonly characterized in terms of phase preference, such description for modulation by SWRs is lacking. To capture the dynamics of AT neurons around time of SWRs, we used jPCA, a method that captures the rotational dynamics of a neuronal population during non-periodic behavior (Churchland, 2012). The projection onto the jPC subspace describes the various temporal responses of a population of neurons in a two-dimensional trajectory during a "pseudo-cycle". Specifically, we determined the jPC basis from the ensemble of z-scored cross-correlograms relative to SWRs (see Methods) and projected each cross-correlogram on the first two jPC components (Fig. 3f). In the two-dimensional jPC projection space, each neuron can be attributed a "phase" that corresponds to the angle from the positive direction on the first jPC axis.

Page 7 4th paragraph – I assume theta refers to REM sleep here

The referee is right. To clarify this point in the main text, we changed the following sentence to mention REM sleep at the beginning of the section:

"The firing of AT neurons relative to REM theta was analyzed, as commonly done, in terms of preferred phase and modulation amplitude."

Reviewer #2 (Remarks to the Author):

In the present manuscript authors report that head-direction (HD) neurons in the anterior thalamus (AT) present a homogeneous subpopulation of AT neurons that can be distinguished from other AT neurons by their intrinsic properties and coupling with population activity in the hippocampus. Specifically, HD cells were clustered in the anterior dorsal (AD) thalamus, exhibited distinctive spike train dynamics (namely, short refractory period and lack of burst firing), their activity was not modulated by hippocampal theta oscillations, but it was tightly coupled with hippocampal sharp-wave ripples (SWRs). About 8% of all recorded neurons in AT were classified as HD cells; the remaining neurons (> 90% of recorded neurons) were assigned to non-HD cells. The non-HD cells presented a heterogeneous population with broadly distributed properties. Their activity was differentially modulated around SWRs; a large proportion of non-HD cells (38%) was phase locked to hippocampal theta oscillations. The difference in spike train dynamics between HD and non-HD cells was also revealed at a slow time scale. Due to specific coupling of HD cells with the hippocampal population activity as revealed by SWRs, authors suggest that this AT cell population plays a key role in limbic offline processing by shaping ensemble replay. Overall, it is an interesting study, the results are novel, and the manuscript is well written. The data were analyzed using conventional, but also a number of refined algorithms. Most importantly, authors describe intrinsic properties of a relatively large population of AT neurons (> 2000) and report substantial diversity between individual AT neurons even within the same anatomically defined

nucleus. In addition, authors linked the intrinsic properties of AT neurons with their coupling with hippocampal population activity (theta and SWRs). Large-scale thalamic recordings in behaving animals remain rare ; therefore, this study is definitely a big and valuable achievement and will present interest to a broader audience.

We would like to thank the referee for this very kind appreciation of our work.

Major concerns:

- Throughout the text, it is not always clear if authors considered awake or sleep ripples or both (except when Up and Down states were identified). Was thalamic modulation different around awake or sleep ripples? Did modulation depend on ripple features as amplitude ?

The referee raises a very important point here. We have only considered sleep ripples for this study. In fact, the detection of awake ripples proved to be challenging in our dataset as the task did not favored long periods of immobility of the animal. Thus, we are not able to isolate a sufficiently large number of awake ripples to answer the referee's question. Nevertheless, it seems that the HD system is anchored to the current heading of the animal, independent of the animal's behavior (running or standing still). This suggests that the thalamic HD signal remains stable around the time of awake SWRs, but we are afraid we cannot conclude about this with the present dataset.

To make it clear, we've added qualified SWRs by writing "NREM sleep SWRs" at various places in the manuscript. We have also added the following sentence in the first paragraph of the discussion:

"One limitation of our study is that we only analyzed NREM sleep SWRs. The behavioural task was not designed to maximize the number of awake SWRs, and further studies are needed to determine the relationship between HD and non-HD neurons of the AT and awake SWRs."

- Figure 1b shows that at time of a given SWR some HD cells were active while others were not. Figure 2d shows that overall all HD cells (n = 161) appear to be SWR-modulated. How consistent was modulation of a given HD cell across successive SWRs? If I understood correctly, authors claim that at time of a given SWR a subset of HD cells with similar orientation would be co-activated. If this is correct, I suppose, authors do not have sufficient amount of simultaneously recorded HD cells to illustrate this point. If such coherent activation of HD cells with the same orientation exists offline, is it hard-wired? This point shall be further clarified and maybe addressed in discussion.

This is an interesting point. The accuracy of decoding (using supervised or unsupervised methods) obviously depend on the number of simultaneously recorded neurons and the distribution of their preferred phase. In the case of HD neurons, good decoding can be achieved with 5-10 neurons (as shown in early studies from the Reddish Lab for example, confirmed in Peyrache et al., 2015). In the present study, we restricted the decoding during SWRs for sessions with at least 10 HD neurons. In addition, Bayesian decoding informs only poorly on the accuracy of the decoding (the posterior probability of decoding is often very high, except during period of low population firing for example). Here, we have used a stricter criterion: we considered only the sessions during which ISOMAP gave a ring-shaped distribution of embedding, confirming that a HD signal can be theoretically decoded for all possible angles. Based on this criterion, one session was excluded from the analysis (see

Supplementary Fig. 1). Thus, we have little doubt that we can accurately decode a virtual direction during SWRs.

- Figure 2d shows that a proportion of non-HD cells were also SWR-modulated? Plotting the distribution of SWR 'energy' for non-HD vs HD cells would be helpful to assess the modulation strength across these two thalamic populations.

We agree with the referee. The distribution of SWR energy for HD and non-HD cells is shown in Figure 3c. The two distributions mostly overlap with stronger modulation for a small proportion of non-HD neurons.

- Have authors attempted to cluster the modulation profiles of non-HD cells? One can suspect a more complex temporal dynamics of modulation profiles of different populations of AT (non-HD) neurons, as seen on Figure 2d. In general, I find averaging across all non-HD cells (> 90% of recorded cells) not justified, if not misleading.

To avoid showing misleading representation of the average activity of non-HD neurons, we show all the data in Figure 2d. While the average SWR modulation of non-HD neurons may appear flat, the individual SWR responses show various modulation profiles, either positively or negatively modulated. In order to gain an accurate representation of the population response that goes beyond the simple average representation, we turned to the jPCA analysis as shown in Figure 3.

- Concerning theta modulation : Do AT neurons (page 7) include HD cells? Was correlation between SWR energy and theta modulation significant when HD cells only were considered?

The group of significantly modulated AT neurons contained 21 HD neurons (out of 767 theta-modulated neurons). SWR energy and theta modulation was significantly correlated for the 161 HD neurons of the data ($r = 0.21$, $p = 0.006$), yet, this correlation is lower than the non-HD neurons ($r = 0.31$, $p < 0.001$).

- Authors' statements about causality of cross-regional interactions are not supposed by presented results and need to be toned down. For example, page 6, last paragraph and also in the discussion.

Following the referee's suggestion, we have toned down the mentioned statements in the last paragraph of the second Results section:

"It can thus be concluded that HD neurons (and, by extension, neurons of the AD nucleus) fire homogeneously around the time of SWRs, pointing in a particular direction. "

Minor

- Abstract : I suggest to revise the sentence to make it more informative "...HD cells ... are set in stable states ... before SWRs ". Authors need to clarify what they mean by "stable states". Same in the last paragraph of Introduction.

We thank the referee for the suggestion. We have updated the abstract with the following sentence:

“Here, by simultaneously recording ensembles of neurons in the anterior thalamus and local field potentials in the CA1 area of the hippocampus, we show that the head-direction (HD) cells of the anterodorsal nucleus are set in stable directions immediately before SWRs.”

And the introduction:

“We show that HD cells are specifically coupled to SWRs, increasing their gain and firing coherently for a particular direction during SWRs.”

- NREM is not always followed by sleep to read NREM sleep.

We have updated the manuscript accordingly.

- Page 2, line 5: revise “...HD cells remain coordinated during sleep...” as it is not clear what authors mean without consulting referenced papers. Same in the second paragraph of Results: “... convey a coherent signal...” phrase not informative without reading referenced papers. Please, revise to make its meaning clearer.

We have now updated the manuscript to clarify the referenced papers:

“The contribution of AD HD cells to offline processing of spatial signals is further demonstrated by the fact that HD cells maintain a coherent firing activity such that a single direction can be decoded at any moment during sleep (Peyrache et al, 2015, Chaudhuri et al, 2019). Additionally, AD HD cells lead their main cortical targets (in the post-subiculum), independently of brain states (Viejo et al, 2018). This suggests that during sleep AD neurons continue conveying a coherent HD signal to the navigation system, including the hippocampus, independent of the current heading of the animal (Peyrache et al, 2019).”

- Figure 2c: why number of SWRs is limited to 100. Are they randomly selected or successive SWRs?

ISOMAP does not scale well with the size of the dataset (greedy in memory). The 2D embedding obtained with ISOMAP around time of SWRs have to be computed with wake data to uncover the ring-shaped topology, certainly because the HD signal drifts much faster during NREM (ideally, one should use very small bins, perhaps as short as 5ms, but then it raises the issue of extremely sparse data with the number of neurons we typically record). More sophisticated methods can extract ring topology from NREM data (as we showed recently in Chaudhuri et al, 2019), however these methods are also computationally very heavy.

We could have run some analyses on large computer clusters but we figured that we obtained very reasonable results by taking the same large chunk of awake data and adding data around 100 SWRs taken randomly, so that each SWR is eventually included. By doing so, the ring-topology was identical at each iteration and neural ‘trajectories’ could be decoded for each SWR.

We have clarified the explanation in the method section:

“To further reduce the computational requirement for long spike trains (i.e. typically more than

30000 time bins), we repeated the ISOMAP projection using a random subset of 150 SWRs and 150 random NREM sleep time bins for baseline quantification with the same 15 min of awake data, until all SWRs were analyzed. This ensures that at each iteration, ISOMAP gives a similar topology of embeddings and that neuronal 'trajectories' around SWRs can be compared to each other for a given session."

Reviewers' Comments:

Reviewer #1:

Remarks to the Author:

The revision has fully addressed my questions and concerns.

Reviewer #2:

Remarks to the Author:

Authors have satisfactorily addressed most of my comments.

Without intending to delay the review process, I would kindly ask the authors to point clearly to me (and potential readers) the result of what analysis indicates signaling the direction specificity during a given SWR by a population of coherently activated HD-cells? Except for examples on Figure 1b, authors present population data averaged over all SWRs. I agree that population result is sufficient to conclude about 'stabilization of HD population' prior SPWs, but does this 'stabilized state' convey direction-selective information at the time of a given SWR or all directions can be theoretically decoded at the time of a given SWR? I only want to clarify the status: is it empirical finding or theoretically possible construct? I do understand that my question about direction specificity may go beyond the scope of the present study, but authors strongly imply on its functional role for off-line processing (replay of spatial trajectories).

Please, ignore my comment if you feel that this point is sufficiently addressed in the manuscript.